# Pollen Paternity Can Affect Kernel Size and Nutritional Composition of Self-Incompatible and New Self-Compatible Almond Cultivars

**Wiebke Kämper** [1,*] **, Grant Thorp** [2] **, Michelle Wirthensohn** [3] **, Peter Brooks** [4] **and Stephen J. Trueman** [1]

1   Environmental Futures Research Institute, School of Environment and Science, Griffith University, Nathan, QLD 4111, Australia; s.trueman@griffith.edu.au
2   Plant & Food Research Australia Pty Ltd., Albert Park, VIC 3206, Australia; grant.thorp@plantandfood.com.au
3   Waite Research Institute, School of Agriculture, Food and Wine, The University of Adelaide, Glen Osmond, SA 5064, Australia; michelle.wirthensohn@adelaide.edu.au
4   Genecology Research Centre, University of the Sunshine Coast, Maroochydore, QLD 4558, Australia; pbrooks@usc.edu.au
*   Correspondence: w.kaemper@griffith.edu.au

**Abstract:** Breeding programs for horticultural tree crops focus on enhancing productivity, including developing tolerance to pests and diseases and improving crop quality. Pollination services are often critical for crop production, and pollen parents can affect crop quality. We often do not know which pollen parents produce highest quality offspring or, in self-compatible cultivars, how much of the crop comes from cross- versus self-pollination. We quantified the proportions of self- and cross-paternity in an open pollination setting of five standard commercial almond cultivars and of six new almond cultivars selected for yield, kernel size, taste or self-compatibility. We assessed how pollination by different parents affected kernel size and nutritional quality. Kernels from most commercial cultivars and from the new cultivars selected for taste and size resulted almost entirely from cross-pollination. Most kernels from the commercial cultivar 'Price' resulted from cross-pollination but 21% resulted from self-pollination. In contrast, 48–91% of kernels from the new self-compatible cultivars resulted from self-pollination. Different cross-pollen parents did not greatly affect kernel size or quality. The proportions of self-paternity in the new self-compatible cultivars varied strongly in an open pollination setting suggesting that some cultivars may be good candidates for establishing monovarietal orchards.

**Keywords:** cross-pollination; fatty acid; kernel quality; nutrients; paternity; self-fertile

## 1. Introduction

Breeding and crop improvement programs aim to enhance yield, quality and genetic diversity of crops [1]. Optimal yield in many horticultural crops depends on successful pollination [2,3]. Self-incompatible plants require cross-pollination to reproduce sexually and some self-compatible plants have increased fruit set when they are cross-pollinated [4–6]. Trees of a single cultivar represent a single clone in clonally-propagated crops such as avocado, citrus and many tree nuts [7]. Therefore, cross-pollination occurs only when the stigma of one cultivar receives pollen from another cultivar, whereas self-pollination occurs when a stigma of one cultivar receives pollen from flowers of the same cultivar [8].

Breeding programs of traditionally self-incompatible plants are focusing on developing self-compatible cultivars that do not depend on synchronised flowering between two closely-interplanted cultivars [9–11]. Studies have investigated fruit set of new self-compatible cultivars of various crops after applying cross- or self-pollen to flowers [12–15]. Some self-compatible plants have increased size and quality when cross-pollinated [4,5,16,17], and so it is important to compare the fruit resulting from self- and

cross-pollination for the new self-compatible cultivars. The effect of different pollen parents on fruit characteristics is a phenomenon termed xenia [18]. Fruit set in self-compatible blueberry cultivars does not differ between self- and cross-pollinated flowers but fruit resulting from cross-pollination and open-pollination are heavier than fruit resulting from self-pollination [19]. We do not know the proportions of self- and cross-pollinated fruit in new self-compatible cultivars in open pollination settings for most crops. We also do not know whether self-pollinated and cross-pollinated fruit of most crops differ in fruit size or nutritional quality.

Almond cultivars are traditionally self-incompatible, which is why orchards are typically established with trees of different cultivars planted in alternate rows [20]. Large numbers of honey bee hives are spread within almond orchards to ensure movement of cross pollen and to achieve cross-pollination. Most orchards consist of several cultivars to improve overlap of flowering between cultivars [11]. Planting alternating rows of different cultivars, however, has drawbacks for orchard management because irrigation, fertiliser application and harvesting could be simplified if large blocks with single cultivars were planted instead [21]. Self-compatible almond cultivars have been developed over the past decades [11,22]. Some studies have found no size or quality differences between self- and cross-pollinated kernels whereas others have found that cross-pollination increases in-shell mass, kernel mass, in-shell volume and kernel volume, and alters oil content and oil composition, compared with those of kernels from self-pollination [23–25]. We know from hand-pollination and bagging experiments that new self-compatible almond cultivars can set self-pollinated and cross-pollinated fruit but it is unclear how much of the crop at harvest results from self- and cross-pollination after open orchard pollination.

We collected fruit from a range of self-incompatible and self-compatible cultivars including five standard commercial cultivars and six new cultivars that were selected for size, taste or self-compatibility. We aimed to assess the levels of self-paternity and cross-paternity in an open pollination experiment. We also aimed to quantify how different paternity affects kernel quality parameters such as size, mineral nutrient concentrations and fatty acid composition. We hypothesised that all commercial cultivars would have a high proportion of cross-pollinated kernels whereas the new self-compatible cultivars would have a high proportion of self-pollinated kernels. We also hypothesised that pollen paternity would affect kernel size and quality.

## 2. Materials and Methods

### 2.1. Study Sites and Design

The study was conducted in two commercial orchards in the Riverland region of Australia. Orchard 1 was near Renmark (34°06′18″ S 140°53′34″ E) in South Australia. Orchard 2 was at Lindsay Point (34°06′12″ S 141°01′37″ E) in Victoria. Orchard 1 was planted with five standard commercial cultivars, 'Carmel', 'Monterey', 'Nonpareil', 'Peerless' and 'Price'. This orchard covered 1129 ha. Tree spacing was 6.5 m between rows and 5.5 m within a row. Trees were planted in 2007 and trunk circumference at the time of the experiment, measured at 10 cm above the graft, was $82.0 \pm 5.1$ cm ($n = 24$). Five European honey bee hives per hectare were placed within the orchard during flowering, with 20 hives every 19 rows. Orchard 2 consisted of three standard commercial cultivars, 'Carmel', 'Nonpareil' and 'Peerless', and 57 novel genotypes including six new cultivars from the Australian almond breeding program [11]. These new cultivars had been selected for kernel size ('Maxima'), taste ('Rhea') or self-compatibility ('Capella', 'Carina', 'Mira' and 'Vela'). Orchard 2 covered 142 ha. Tree spacing was 7.3 m between rows and 4.0 m within a row. Trees were planted in 2013, apart from 'Vela', which was planted in 2010. Trunk circumference at the time of the experiment was $57.9 \pm 9.3$ cm ($n = 48$). Seven European honey bee hives per hectare, with 20 hives every 10 rows, were placed within this orchard during flowering. Trees were planted in single rows of each cultivar in both orchards, with 'Nonpareil' trees planted in every second row and multiple other cultivars in the other rows.

The numbers of honey bees that contacted a flower within a 5-min period were counted in a 1 m$^3$ quadrat on the illuminated side of the tree in both the morning (800–1130 h) and the afternoon (1230–1600 h) of three days during peak flowering for seven cultivars. These counts were performed on 13, 14 and 16 August 2019 at Orchard 1 and on 12, 14 and 16 August 2019 at Orchard 2.

At Orchard 1, fruit were collected from four standard commercial cultivars ('Carmel', 'Monterey', 'Nonpareil' and 'Price'). At Orchard 2, fruit were collected from two commercial cultivars ('Nonpareil' and 'Peerless') and the six new cultivars from the Australian almond breeding program. Mature fruit were sampled from six trees in a single row of each cultivar in February or March 2020. The first five trees at the start of a row were avoided, and then every fifth or tenth tree was sampled, depending on the row length. Sixteen fruit were harvested from each tree in a stratified design, either from the orchard floor if mechanical shaking had already occurred or from the tree canopy. Each tree (or orchard floor under the canopy) was divided into four quadrants on each side of the tree for the stratified sampling, and two fruit were sampled per quadrant, one from the inside and one from the outside of the canopy. We also collected a leaf sample from each tree for genotyping to confirm its cultivar identity.

Ten fruit per tree were then selected randomly for further processing. Kernel moisture content of the other six fruit per tree was assessed in a laboratory oven, with kernels from all cultivars having moisture content ≤5% except for 'Price', which had 16% moisture content. The ten fruit per tree were hulled, the in-shell mass of each nut was recorded before cracking, and then kernel mass was recorded. Shelling percentage was calculated as the percentage of in-shell mass comprised of kernel. Individual kernels were placed in a plastic bag and crushed with a mortar and pestle. Three subsamples were taken, consisting of (a) ~50 mg to assess the genotype and identify the pollen parent; (b) ~200 mg to determine mineral nutrient concentrations; and (c) the remainder to assess fatty acid composition. All kernels were used for genotyping, but only a selection that represented the most common parents for each cultivar was used for mineral nutrient and fatty acid analysis.

### 2.2. Paternity Testing

DNA extraction followed the protocol for glass-fibre plate DNA extraction for plants (http://ccdb.ca/resources/) developed by Ivanova, Fazekas [26]. We used disposable 2.3 mm and 0.1 mm zirconia/silica beads prior to shaking on a TissueLyser II (Qiagen, Hilden, Germany) [27]. The DNA of each sample was amplified at 10 microsatellite loci suitable for building a reference database of the four (Orchard 1) and 60 (Orchard 2) potential pollen parents in at least a 1.5 km radius from our sampling location (Table 1) [28]. The 5′ end of each forward primer was fluorescently labelled (Table 1). Three multiplex PCRs were performed per sample using the Qiagen Type-it Microsatellite PCR Kit (Qiagen, Hilden, Germany). Reactions were performed in 12.5 μL reaction volumes containing approximately 20 ng DNA template, 5.6 μL Type-it Multiplex PCR Master Mix, 2 μM of each primer and 3.6 μL RNase-free water. PCR was performed with initial denaturation at 95 °C for 5 min, followed by 32 cycles of 95 °C for 30 s, 57 °C for 90 s, and 72 °C for 30 s, followed by final elongation at 60 °C for 30 min.

Genotypes were generated using an AB 3500 Genetic Analyser (Applied Biosystems, Foster City, CA, USA) and allele sizes scored relative to an internal standard (600 LIZ® Size Standard, Applied Biosystems, Foster City, CA, USA) using the program GeneMarker version 2.6.3 (SoftGenetics, State College, PA, USA). The pollen parent of each kernel was assigned using the software CERVUS version 3.0.7 [29]. Simulations on paternity were run using the following parameters: 100,000 simulated offspring, the proportion of mistyped loci was set at 0.01, and the proportion of candidate fathers sampled was estimated at 0.99. CERVUS calculated a likelihood ratio representing how much more likely a putative pollen parent was compared with a pollen parent selected randomly from the reference library. The likelihood ratio was expressed as the LOD score, which is the natural logarithm of the likelihood ratio. We only considered pollen parents possible if CERVUS assigned a positive

LOD score. We only assigned a specific pollen parent if Cervus assigned a positive LOD score and the best match met the 95% strict-confidence level (based on 100,000 iterations).

**Table 1.** Characterisation of ten polymorphic microsatellite loci used to determine paternity of almond kernels (Fernández i Martí et al. 2009).

| Locus | Primer Sequences (5' to 3') | Fluorescent Label | $T_a$ | Allele Sizes |
|---|---|---|---|---|
| BPPCT001 | F: AATTCCCAAAGGATGTGTATGAG | NED | 57 | 121–178 |
| | R: CAGGTGAATGAGCCAAAGC | | | |
| BPPCT007 | F: TCATTGCTCGTCATCAGC | NED | 57 | 125–162 |
| | R: CAGATTTCTGAAGTTAGCGGTA | | | |
| BPPCT025 | F: TCCTGCGTAGAAGAAGGTAGC | FAM | 57 | 156–193 |
| | R: CGACATAAAGTCCAAATGGC | | | |
| CPDCT025 | F: GACCTCATCAGCATCACCAA | PET | 62 | 156–193 |
| | R: TTCCCTAACGTCCCTGACAC | | | |
| CPDCT045 | F: TGTGGATCAAGAAAGAGAACCA | NED | 62 | 132–181 |
| | R: AGGTGTGCTTGCACATGTTT | | | |
| CPPCT006 | F: AATTAACTCCAACAGCTCCA | FAM | 59 | 156–216 |
| | R: ATGGTTGCTTAATTCAATGG | | | |
| CPPCT022 | F: CAATTAGCTAGAGAGAATTATTG | VIC | 50 | 221–260 |
| | R: GACAAGAAGCAAGTAGTTTG | | | |
| CPPCT044 | F: TTCTCTTTGGCGTATCAAGGA | FAM | 58 | 153–200 |
| | R: GGTCCCATATCAGCTGAACC | | | |
| CPSCT012 | F: ACGGGAGACTTTCCCAGAAG | NED | 62 | 145–183 |
| | R: CTTCTCGTTTCCTCCCTCCT | | | |
| PMS40 | F: TCACTTTCGTCCATTTTCCC | VIC | 55 | 88–135 |
| | R: TCATTTTGGTCTTTGACCTCG | | | |

Ta: Annealing temperature.

### 2.3. Mineral Nutrient Concentrations and Fatty Acid Composition

We determined nutrient concentrations and fatty acid composition from a subset of 234 kernels representing the two most common cross-pollen parents for three self-incompatible cultivars and a subset of self-pollinated and cross-pollinated kernels from two self-compatible cultivars. Nitrogen (N) and carbon (C) concentrations were determined by combustion analysis (TruSpec®, LECO Corporation, St. Joseph, MI, USA) [30,31]. Aluminium (Al), boron (B), calcium (Ca), copper (Cu), iron (Fe), magnesium (Mg), manganese (Mn), potassium (K), phosphorus (P), sodium (Na), sulphur (S) and zinc (Zn) concentrations were determined by inductively coupled plasma–atomic emission spectroscopy (Vista Pro®, Varian Incorporation, Palo Alto, CA, USA) after nitric and perchloric acid digestion [32,33].

Oil extraction and fatty acid analysis followed the protocol described by Bai, Brooks [34]. Each subsample was finely crushed and added to approximately 20 mL of pentane before being stirred for 15 min. The pentane extract was transferred into a round-bottom flask and pentane evaporated for 5 min using an air-tight vacuum rotator. A total of 0.7 mL of anhydrous methanol dibutylhydroxytoluene solution and 25 μL of 32% HCl was added to 1 μL of extracted oil. The mixture was incubated for 20 h at 65 °C before adding 0.5 mL of hexane and 0.5 mL of deionised water, then shaking for 30 s, before rinsing again with 0.5 mL deionised water. The top layer was collected and $Na_2SO_4$ added to remove water from the oil in hexane. Fatty acid composition was determined by gas chromatography–mass spectrometry. Peak areas were used to calculate contents of each fatty acid.

### 2.4. Statistical Analysis

We calculated the percentages of cross-pollinated kernels, self-pollinated kernels and kernels with unassigned parentage for each tree, and calculated mean percentages for each cultivar per site. We compared kernel size, shelling percentage, kernel shape, and nutritional quality (mineral nutrient concentrations and fatty acid composition) between different cross-pollen parents for three self-incompatible cultivars and between self-pollinated and cross-pollinated kernels for two self-compatible cultivars. Cultivars were selected to avoid very small sample sizes, based on having at least two pollen parents with a minimum of ten kernels. Specifically, we compared (i) in-shell mass, (ii) kernel mass, (iii) shelling percentage, (iv) kernel shape (length:width ratio), (v) concentrations of each of 14 nutrients, (vi) relative contributions of 5–6 acids to the total fatty acid composition, and (vi) relative contributions of saturated and unsaturated fatty acids to the total fatty acid composition, using linear mixed models with pollen source as a fixed and categorical variable and tree identity as a random effect. Tukey's HSD tests were performed when differences were detected. Kernels with unassigned parentage were removed before statistical analyses. Statistical analyses were performed using R version 3.6.2 [35]. Mixed models were performed with the package 'lme4' and 'multcomp' [36,37].

## 3. Results

Honey bees were active in both orchards, with 4–11 bees observed in each 1 m$^3$ quadrat in a 5-min window during peak flowering (Table 2).

**Table 2.** Number of honey bee visits to almond flowers within a 5-min period on three days during peak flowering.

| Cultivar | Orchard | Number of Visitors |
|:---:|:---:|:---:|
| 'Monterey' | Orchard 1 | $4.8 \pm 1.4$ |
| 'Nonpareil' | Orchard 1 | $3.9 \pm 1.0$ |
| 'Capella' | Orchard 2 | $4.0 \pm 1.1$ |
| 'Carina' | Orchard 2 | $11.3 \pm 2.9$ |
| 'Peerless' | Orchard 2 | $5.2 \pm 2.4$ |
| 'Rhea' | Orchard 2 | $7.4 \pm 1.8$ |
| 'Vela' | Orchard 2 | $10.2 \pm 2.9$ |

Means are presented with SE ($n = 18$).

### 3.1. Cross- and Self-Paternity

At least 83–98% of kernels of the standard commercial cultivars 'Carmel', 'Monterey', 'Nonpareil' and 'Peerless' resulted from cross-pollination and as few as 0–6% resulted from self-pollination (Figure 1). The remaining kernels (0–17%) could not be assigned as either self- or cross-pollinated at a 95% level of confidence (Figure 1). Similarly, 92–94% of kernels of the new cultivars selected for kernel size ('Maxima') and taste ('Rhea') resulted from cross-pollination and as few as 2% resulted from self-pollination (Figure 1). In contrast, only 5–42% of kernels of the new self-compatible cultivars 'Capella', 'Carina', 'Mira' and 'Vela' resulted from cross-pollination and as many as 48–91% resulted from self-pollination (Figure 1). Again, the remaining kernels (0–10%) could not be assigned as either self- or cross-pollinated (Figure 1). The commercial cultivar 'Price' had substantial numbers of both cross- and self-pollinated kernels, with 76% resulting from cross-pollination and 21% resulting from self-pollination (Figure 1).

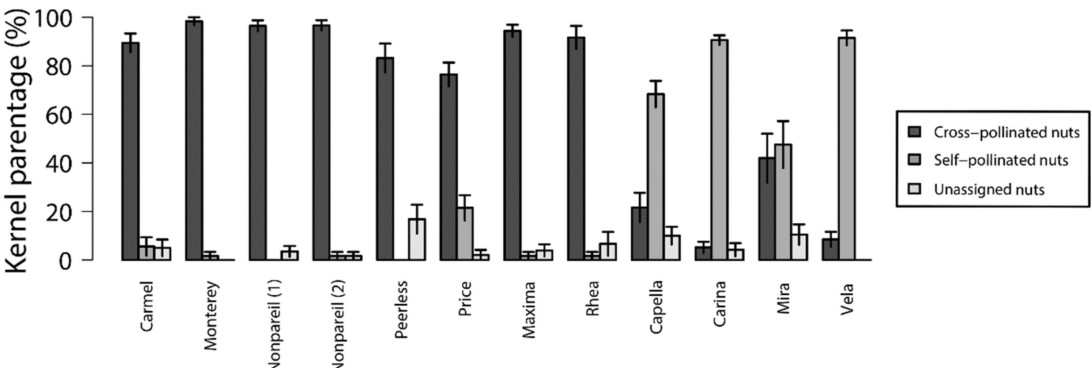

**Figure 1.** The percentage (means ± SE) of cross-pollinated, self-pollinated and unassigned kernels from five standard commercial almond cultivars ('Carmel', 'Monterey', 'Nonpareil', 'Peerless' and 'Price') and six new almond cultivars. New cultivars were selected for size ('Maxima'), taste ('Rhea') or self-compatibility ('Capella', 'Carina', 'Mira' and 'Vela'). 'Nonpareil' kernels were collected from both Orchard 1 (1) and Orchard 2 (2).

*3.2. Effect of Pollen Parentage on Kernel Size, Mineral Nutrient Concentrations and Fatty Acid Composition*

In-shell mass and kernel mass were generally not affected greatly by pollen parentage, except for kernel mass in the commercial cultivar 'Carmel', which was 6% higher in fruit cross-pollinated by 'Price' than by 'Nonpareil' (Table 3), and in-shell mass in the new 'self-compatible' cultivar 'Mira', which was 8% higher in cross-pollinated than in self-pollinated fruit (Table 4). Shelling percentage was not affected by pollen parentage (Tables 3 and 4). Kernel shape, measured as the ratio of kernel height to width, was generally not affected by pollen parentage, except in 'Mira' (Tables 3 and 4). Cross-pollinated kernels in 'Mira' had a higher length to width ratio than self-pollinated kernels (Table 4).

Cross-pollen paternity did not significantly affect the concentrations of nitrogen, boron, copper, phosphorus, potassium, sodium and zinc in almond kernels (Table 3). 'Monterey' kernels pollinated by 'Carmel' had higher carbon concentrations and lower manganese and sulphur concentrations than 'Monterey' kernels pollinated by 'Nonpareil'. 'Nonpareil' kernels pollinated by 'Carmel' had higher carbon concentrations but lower aluminium, calcium, iron, magnesium and manganese concentrations than kernels pollinated by 'Price'. Cross-paternity with 'Carmel' of both 'Monterey' and 'Nonpareil' kernels increased carbon concentrations but decreased other nutrient concentrations compared with the alternative cross-pollen parent (Table 3). These differences were most pronounced for aluminium, iron and manganese, with 'Nonpareil' kernels pollinated by 'Carmel' having 74% lower aluminium, 20% lower iron, and 15% lower manganese than kernels pollinated by 'Price'. Manganese concentrations were also 21% lower in 'Monterey' kernels pollinated by 'Carmel' than by 'Nonpareil'. Cross-pollen paternity did not affect the fatty acid composition of almond kernels greatly, but 'Nonpareil' kernels had slightly higher levels of the monounsaturated fatty acid, palmitoleic acid, when pollinated by 'Carmel' than by 'Price' (Table 3).

Self- and cross-pollinated kernels did not differ in the concentrations of nitrogen, aluminium, boron, calcium, copper, iron, magnesium, sodium, phosphorus, sulphur or zinc (Table 4). Cross-pollinated kernels of two self-compatible cultivars, 'Capella' and 'Mira', had higher carbon concentrations and either lower manganese or lower potassium concentration than self-pollinated kernels (Table 4). Cross-pollinated kernels of 'Capella' had 17% higher levels of the saturated acid, palmitic acid, than self-pollinated kernels whereas the opposite was observed for 'Mira', where cross-pollinated kernels had 8% lower levels of palmitic acid than self-pollinated kernels (Table 4). Cross-pollinated 'Mira' kernels also had a slightly higher ratio of unsaturated to saturated fatty acids than did self-pollinated kernels (Table 4).

**Table 3.** Kernel size, shape, shelling percentage, mineral nutrient concentrations and relative abundances of fatty acids of the two most abundant cross-pollen parents in three commercial cross-pollinated almond cultivars.

| Cultivar | 'Carmel' | | | | 'Monterey' | | | | 'Nonpareil' (Orchard 1) | | | |
|---|---|---|---|---|---|---|---|---|---|---|---|---|
| **Pollen Parent**<br>**Sample Number** | **'Nonpareil'**<br>*n* = 19 | | **'Price'**<br>*n* = 14 | | **'Carmel'**<br>*n* = 16 | | **'Nonpareil'**<br>*n* = 33 | | **'Carmel'**<br>*n* = 41 | | **'Price'**<br>*n* = 10 | |
| In-shell mass (g) | 1.74 ± 0.04 | | 1.82 ± 0.03 | | 2.56 ± 0.06 | | 2.41 ± 0.02 | | 1.99 ± 0.07 | | 1.91 ± 0.08 | |
| Kernel mass (g) | 1.13 ± 0.03 | b | 1.20 ± 0.03 | a | 1.44 ± 0.05 | | 1.38 ± 0.01 | | 1.34 ± 0.04 | | 1.32 ± 0.05 | |
| Kernel length:width ratio | 1.76 ± 0.02 | | 1.77 ± 0.02 | | 1.98 ± 0.03 | | 1.92 ± 0.02 | | 1.67 ± 0.02 | | 1.66 ± 0.02 | |
| Shelling percentage | 65.39 ± 0.60 | | 65.90 ± 0.95 | | 56.34 ± 1.22 | | 57.51 ± 0.83 | | 67.75 ± 0.82 | | 69.29 ± 1.74 | |
| C–Carbon (%) | 62.51 ± 0.13 | | 62.89 ± 0.15 | | 63.27 ± 0.45 | a | 61.99 ± 0.31 | b | 62.69 ± 0.38 | a | 61.11 ± 0.46 | b |
| N–Nitrogen (%) | 3.52 ± 0.12 | | 3.61 ± 0.04 | | 3.29 ± 0.14 | | 3.51 ± 0.11 | | 3.47 ± 0.08 | | 3.67 ± 0.08 | |
| Al–Aluminium (mg/kg) | 5.47 ± 0.22 | | 5.19 ± 0.27 | | 4.35 ± 0.35 | | 4.87 ± 0.31 | | 4.60 ± 0.31 | b | 17.46 ± 9.11 | a |
| B–Boron (mg/kg) | 34.67 ± 3.47 | | 33.99 ± 1.99 | | 40.00 ± 1.18 | | 43.28 ± 1.29 | | 27.23 ± 1.17 | | 30.56 ± 1.95 | |
| Ca–Calcium (mg/kg) | 0.26 ± 0.01 | | 0.24 ± 0.01 | | 0.23 ± 0.03 | | 0.26 ± 0.01 | | 0.19 ± 0.01 | b | 0.22 ± 0.02 | a |
| Cu–Copper (mg/kg) | 7.62 ± 0.70 | | 7.13 ± 0.16 | | 7.32 ± 0.44 | | 7.40 ± 0.33 | | 8.73 ± 0.18 | | 9.38 ± 0.22 | |
| Fe–Iron (mg/kg) | 29.58 ± 1.46 | | 28.77 ± 1.94 | | 33.15 ± 1.61 | | 33.60 ± 1.98 | | 33.39 ± 1.27 | b | 41.83 ± 2.26 | a |
| K–Potassium (mg/kg) | 0.72 ± 0.02 | | 0.75 ± 0.04 | | 0.70 ± 0.03 | | 0.70 ± 0.02 | | 0.69 ± 0.01 | | 0.69 ± 0.01 | |
| Mg–Magnesium (mg/kg) | 0.25 ± 0.01 | | 0.26 ± 0.01 | | 0.26 ± 0.004 | | 0.28 ± 0.01 | | 0.26 ± 0.002 | b | 0.27 ± 0.01 | a |
| Mn–Manganese (mg/kg) | 31.13 ± 1.86 | | 28.21 ± 1.69 | | 23.66 ± 2.07 | b | 29.96 ± 2.15 | a | 29.11 ± 1.90 | b | 34.30 ± 2.74 | a |
| Na–Sodium (mg/kg) | 18.09 ± 0.84 | | 18.96 ± 1.47 | | 17.07 ± 0.81 | | 19.42 ± 1.61 | | 10.47 ± 2.53 | | 14.38 ± 4.94 | |
| P–Phosphorus (mg/kg) | 0.47 ± 0.01 | | 0.50 ± 0.02 | | 0.47 ± 0.01 | | 0.47 ± 0.004 | | 0.47 ± 0.01 | | 0.49 ± 0.02 | |
| S–Sulphur (mg/kg) | 0.09 ± 0.004 | | 0.08 ± 0.003 | | 0.10 ± 0.003 | b | 0.12 ± 0.004 | a | 0.13 ± 0.01 | | 0.13 ± 0.01 | |
| Zn–Zinc (mg/kg) | 24.27 ± 0.55 | | 25.09 ± 1.32 | | 23.94 ± 1.34 | | 24.01 ± 0.82 | | 28.44 ± 0.58 | | 29.38 ± 1.33 | |
| Palmitic acid-C16:0 (%) | 4.76 ± 0.11 | | 4.56 ± 0.13 | | 5.17 ± 0.08 | | 5.19 ± 0.11 | | 5.28 ± 0.10 | | 4.94 ± 0.18 | |
| Palmitoleic acid-C16:1 *cis* (%) | 0.08 ± 0.004 | | 0.06 ± 0.003 | | 0.08 ± 0.02 | | 0.09 ± 0.01 | | 0.14 ± 0.01 | a | 0.11 ± 0.01 | b |
| Stearic acid-C18:0 (%) | 0.68 ± 0.07 | | 0.61 ± 0.04 | | 0.89 ± 0.06 | | 0.79 ± 0.04 | | 0.79 ± 0.04 | | 0.75 ± 0.05 | |
| Oleic acid-C18:1 *cis* (%) | 67.77 ± 1.32 | | 66.63 ± 1.24 | | 70.59 ± 0.90 | | 70.99 ± 0.98 | | 72.65 ± 0.97 | | 74.83 ± 0.80 | |
| Elaidic acid-C18:1 *trans* (%) | – | | – | | – | | – | | – | | – | |

**Table 3.** *Cont.*

| Cultivar | 'Carmel' | | 'Monterey' | | 'Nonpareil' (Orchard 1) | |
|---|---|---|---|---|---|---|
| **Pollen Parent Sample Number** | **'Nonpareil'** *n* = 19 | **'Price'** *n* = 14 | **'Carmel'** *n* = 16 | **'Nonpareil'** *n* = 33 | **'Carmel'** *n* = 41 | **'Price'** *n* = 10 |
| Linoleic acid-C18:2 (%) | 26.71 ± 1.22 | 28.13 ± 1.19 | 23.27 ± 0.89 | 22.94 ± 0.92 | 21.14 ± 0.88 | 19.38 ± 0.87 |
| Saturated fatty acids (SFA) | 5.44 ± 0.14 | 5.18 ± 0.15 | 6.06 ± 0.06 | 5.98 ± 0.13 | 6.07 ± 0.11 | 5.69 ± 0.21 |
| Unsaturated fatty acids (UFA) | 94.56 ± 0.14 | 94.82 ± 0.15 | 93.94 ± 0.06 | 94.02 ± 0.13 | 93.93 ± 0.11 | 94.31 ± 0.21 |
| UFA:SFA | 17.74 ± 0.52 | 18.46 ± 0.55 | 15.64 ± 0.20 | 16.02 ± 0.36 | 15.60 ± 0.27 | 16.78 ± 0.70 |

Means ± SE with different letters within a cultivar are significantly different (mixed model; *n* = 6 trees).

**Table 4.** Size, shape, shelling percentage, mineral nutrient concentrations and relative abundances of fatty acids of cross-pollinated and self-pollinated kernels of two new self-compatible almond cultivars.

| Cultivar | 'Capella' | | | | 'Mira' | | | |
|---|---|---|---|---|---|---|---|---|
| **Pollen Parent Sample Number** | **Cross** *n* = 13 | | **Self** *n* = 41 | | **Cross** *n* = 20 | | **Self** *n* = 27 | |
| In-shell mass (g) | 4.53 ± 0.60 | | 4.13 ± 0.30 | | 3.98 ± 0.30 | a | 3.66 ± 0.30 | b |
| Kernel mass (g) | 1.38 ± 0.06 | | 1.28 ± 0.03 | | 1.57 ± 0.04 | | 1.47 ± 0.02 | |
| Kernel length:width ratio | 1.46 ± 0.01 | | 1.44 ± 0.01 | | 1.53 ± 0.01 | b | 1.58 ± 0.02 | a |
| Shelling percentage | 30.81 ± 0.97 | | 31.30 ± 0.95 | | 39.63 ± 0.86 | | 40.48 ± 0.62 | |
| C–Carbon (%) | 61.41 ± 0.22 | a | 60.08 ± 0.39 | b | 64.30 ± 0.14 | a | 63.31 ± 0.33 | b |
| N–Nitrogen (%) | 4.24 ± 0.12 | | 4.31 ± 0.13 | | 4.08 ± 0.05 | | 4.09 ± 0.03 | |
| Al–Aluminium (mg/kg) | 3.87 ± 0.30 | | 5.14 ± 0.42 | | 7.09 ± 2.26 | | 4.75 ± 0.14 | |
| B–Boron (mg/kg) | 21.02 ± 0.69 | | 21.39 ± 0.47 | | 21.94 ± 1.58 | | 21.63 ± 1.25 | |
| Ca–Calcium (mg/kg) | 0.23 ± 0.02 | | 0.26 ± 0.01 | | 0.23 ± 0.01 | | 0.24 ± 0.02 | |
| Cu–Copper (mg/kg) | 8.84 ± 0.51 | | 9.13 ± 0.61 | | 25.59 ± 12.47 | | 10.97 ± 0.44 | |
| Fe–Iron (mg/kg) | 36.38 ± 1.57 | | 36.62 ± 1.23 | | 43.38 ± 2.33 | | 46.52 ± 1.83 | |
| K–Potassium (mg/kg) | 0.90 ± 0.01 | | 0.94 ± 0.02 | | 0.73 ± 0.02 | b | 0.83 ± 0.04 | a |
| Mg–Magnesium (mg/kg) | 0.27 ± 0.01 | | 0.27 ± 0.003 | | 0.24 ± 0.003 | | 0.25 ± 0.01 | |

**Table 4.** *Cont.*

| Cultivar | 'Capella' | | | | 'Mira' | | | |
|---|---|---|---|---|---|---|---|---|
| Pollen Parent<br>Sample Number | Cross<br>n = 13 | | Self<br>n = 41 | | Cross<br>n = 20 | | Self<br>n = 27 | |
| Mn–Manganese (mg/kg) | 30.50 ± 3.01 | b | 37.46 ± 1.71 | a | 31.88 ± 2.22 | | 39.09 ± 3.74 | |
| Na–Sodium (mg/kg) | 20.20 ± 1.36 | | 39.53 ± 16.78 | | 19.64 ± 1.40 | | 20.77 ± 0.88 | |
| P–Phosphorus (mg/kg) | 0.58 ± 0.01 | | 0.57 ± 0.01 | | 0.49 ± 0.01 | | 0.54 ± 0.02 | |
| S–Sulphur (mg/kg) | 0.16 ± 0.01 | | 0.16 ± 0.01 | | 0.14 ± 0.01 | | 0.14 ± 0.01 | |
| Zn–Zinc (mg/kg) | 38.64 ± 1.64 | | 40.83 ± 1.07 | | 40.25 ± 1.83 | | 44.60 ± 0.77 | |
| Palmitic-C16:0 (%) | 3.77 ± 0.14 | a | 3.22 ± 0.19 | b | 5.73 ± 0.11 | b | 6.28 ± 0.04 | a |
| Palmitoleic-C16:1 *cis* (%) | 0.08 ± 0.01 | | 0.08 ± 0.004 | | 0.22 ± 0.02 | | 0.24 ± 0.02 | |
| Stearic-C18:0 (%) | 0.98 ± 0.12 | | 1.22 ± 0.07 | | 0.81 ± 0.03 | | 0.74 ± 0.04 | |
| Oleic-C18:1 *cis* (%) | 83.63 ± 0.88 | | 85.81 ± 1.02 | | 71.64 ± 0.79 | | 70.82 ± 0.72 | |
| Elaidic-C18:1 *trans* (%) | – | | – | | 2.05 ± 0.05 | | 2.04 ± 0.09 | |
| Linoleic-C18:2 (%) | 11.54 ± 0.77 | | 9.68 ± 0.82 | | 19.55 ± 0.65 | | 19.86 ± 0.78 | |
| Saturated fatty acids (SFA) | 4.76 ± 0.23 | | 4.44 ± 0.23 | | 6.54 ± 0.11 | b | 7.03 ± 0.06 | a |
| Unsaturated fatty acids (UFA) | 95.24 ± 0.23 | | 95.56 ± 0.23 | | 93.46 ± 0.11 | a | 92.97 ± 0.06 | b |
| UFA:SFA | 20.32 ± 1.10 | | 22.24 ± 1.22 | | 14.36 ± 0.25 | a | 13.30 ± 0.12 | b |

Means ± SE with different letters within a cultivar are significantly different (mixed model; *n* = 6 trees).

## 4. Discussion

Kernels of most standard commercial almond cultivars resulted almost entirely from cross-pollination, although 21% of kernels of the commercial cultivar 'Price' were self-pollinated. In contrast, 48–91% of kernels of the new self-compatible cultivars resulted from self-pollination. Pollen paternity did not greatly affect fruit size or mineral nutrient concentrations of almond kernels. However, kernels of commercial cultivars when cross-pollinated by 'Carmel' had slightly higher carbon concentrations and slightly lower concentrations of some other nutrients. Cross-pollinated kernels of two self-compatible cultivars had higher carbon concentrations and either lower potassium or lower manganese concentrations than self-pollinated kernels. Pollen paternity also did not greatly affect fatty acid composition, although 'Nonpareil' kernels had slightly higher abundance of a monounsaturated fatty acid, palmitoleic acid, when pollinated by 'Carmel' than by 'Price'. Cross-pollination increased palmitic acid abundance in 'Mira' kernels and contributed to a slightly higher ratio of unsaturated to saturated fatty acids. In contrast, cross-pollination decreased palmitic acid abundance in 'Capella' kernels.

Our data confirm that trees of 'Carmel', 'Monterey', 'Nonpareil', 'Peerless' and, to a lesser extent 'Price', rely on cross-pollination. 'Maxima' and 'Rhea' also relied strongly on cross-pollination in an open-pollination orchard setting. Our results demonstrate that the four new self-compatible cultivars produced kernels readily by self-pollination in an open-pollination setting. Previous studies have compared fruit set of self-compatible, autogamous almond cultivars in controlled experiments after bagging branches and applying cross- or self-pollen [23,38] but could not draw conclusions about self-paternity and cross-paternity rates in an open pollination setting. Our results show that 'Capella, 'Carina', 'Mira' and 'Vela' self-pollinate in an open pollination setting. Our results also demonstrate high levels of cross-paternity in many cultivars, suggesting that cross-pollen was moved across the orchard. Together these observations suggest that the four self-compatible cultivars self-pollinated effectively even though both self- and cross-pollen were likely deposited on stigmas within this multi-cultivar orchard. High levels of self-paternity in an open pollination setting, as observed in 'Carina' and 'Vela', suggest that monovarietal orchards could be established successfully. However, it will be important to compare yields between the commercial and new cultivars. It will also be important to investigate whether self-compatible and self-incompatible cultivars require the same number of floral visits for optimal fruit production. Self-compatible blueberry cultivars have been found to require fewer insect visits than partially self-incompatible cultivars [19].

In contrast to some other self-compatible cultivars in this study, the self-compatible cultivar 'Mira' carried a similar number of cross-pollinated and self-pollinated kernels. We had hypothesised that self-compatible cultivars would carry mostly self-pollinated fruit because most pollen deposited on stigmas would be self- rather than cross-pollen. We considered self-pollen more likely to be deposited than cross-pollen because (i) trees were planted in single rows of each cultivar in both orchards, and so the closest neighbouring tree was a source of self-pollen; (ii) in other crops, honey bees often forage on trees along the same orchard row rather than across rows, suggesting that honey bees were more likely to carry and deposit self-pollen than cross-pollen [39–41]; and (iii) flowering between trees of the same cultivar may be more synchronised than flowering between different cultivars. Selective abortion of self-pollinated fruitlets or differences in pollen tube growth after self-pollination and cross-pollination could explain the high numbers of cross-pollinated fruit in 'Mira' [42,43]. No differences in the number of pollen tubes reaching the ovary or in fruit set have been found after self- versus cross-pollination of six other self-compatible almond cultivars [23]. However, in those laboratory-based experiments, branches received either self- or cross-pollen and thus carried either self-pollinated or cross-pollinated fruitlets. In our field experiment, trees potentially carried both self- and cross-pollinated fruitlets, creating an opportunity to selectively abort self-pollinated fruitlets.

Differences in size between kernels pollinated by different cross-pollen parents were small but in cultivars 'Nonpareil' and 'Monterey', the cross-pollen parent affected some

of the mineral nutrient concentrations. Aluminium, iron and manganese concentrations were significantly lower when 'Nonpareil' kernels were pollinated by 'Carmel' than by 'Price', and manganese concentrations were significantly lower when 'Monterey' kernels were pollinated by 'Carmel' than by 'Nonpareil'. Xenia effects of different cross-pollen parents on fruit size have been observed previously in almond cultivars 'Shahrood 12' and 'Shahrood 21' [44], but nutritional quality was not investigated. The slightly higher concentrations of some mineral nutrients of the industry-preferred 'Nonpareil' kernels when pollinated by 'Price' than by 'Carmel' may be significant for the almond industry [45].

Cross-pollinated 'Mira' kernels had higher in-shell mass and lower length:width ratio than self-pollinated 'Mira' kernels. Cross-pollinated 'Mira' kernels also had higher concentrations of carbon but lower concentrations of potassium, and lower abundance of the saturated palmitic acid, contributing to a slightly higher ratio of unsaturated to saturated fatty acids. Other studies have found that self-pollinated kernels have more δ-tocopherol and a higher oleic:linoleic acid ratio than cross-pollinated kernels in some self-compatible almond cultivars [25,46]. Differences in this study were larger than the observed differences between different cross-pollen parents in the commercial cultivars, and between cross-pollinated and self-pollinated kernels in the new self-compatible cultivars. The increased in-shell mass and higher ratio of unsaturated to saturated fatty acids of cross-pollinated 'Mira' kernels could be more desirable for the almond industry.

Breeding and crop improvement programs aim to enhance quality and performance of crops [1]. Understanding how pollen parentage affects kernel quality of cultivars provides important information to assist genotype selection and to plan new orchards with ideal cultivar combinations. The commercial cultivar 'Carmel' and the new self-compatible cultivar 'Capella', for example, have higher ratios of unsaturated to saturated fatty acids than the commercial cultivars 'Nonpareil' and 'Monterey' and the self-compatible cultivar 'Mira'. This higher ratio could be beneficial for human health because a diet rich in unsaturated fatty acids decreases LDL-cholesterol levels and other cardiovascular risk factors [47,48]. Investigating performance of self-compatible cultivars and understanding how likely a self-compatible cultivar is to produce fruit with self- or cross-paternity in an open-pollination setting provides important insights into whether monovarietal orchards of these cultivars could produce good yields.

## 5. Conclusions

Our results demonstrate that the four new self-compatible almond cultivars are setting self-pollinated fruit in an open pollination setting. Interestingly, though, the proportions of self-paternity among fruit differed between the self-compatible cultivars, ranging from 48 to 91%. Xenia effects on fruit size and quality of some pollen parents were observed in both the commercial and new cultivars, but differences in fruit size and quality between cultivars were often larger than differences within cultivars. Monovarietal orchards would have synchronised flowering between trees and would simplify orchard management, but we still need to better understand the yield and quality of the new self-compatible cultivars compared with long-standing and better-known commercial cultivars.

**Author Contributions:** Conceptualization, W.K. and S.J.T.; methodology, W.K.; software, W.K., P.B., S.J.T.; validation, W.K., G.T., M.W., P.B., S.J.T..; formal analysis, W.K.; investigation, W.K., G.T., M.W., P.B., S.J.T.; resources, W.K., G.T., M.W., P.B., S.J.T.; data curation, W.K.; writing—original draft preparation, W.K.; writing—review and editing, W.K., G.T., M.W., P.B., S.J.T.; visualization, W.K.; supervision, S.J.T.; project administration, S.J.T.; funding acquisition, S.J.T. All authors have read and agreed to the published version of the manuscript.

**Funding:** This project was funded by Project PH16001 of the Hort Frontiers Pollination Fund, part of the Hort Frontiers strategic partnership initiative developed by Hort Innovation, with co-investment from Griffith University, University of the Sunshine Coast, Plant & Food Research and contributions from the Australian Government.

**Institutional Review Board Statement:** Not applicable.

**Informed Consent Statement:** Not applicable.

**Data Availability Statement:** The data presented in this study are available on request from the corresponding author. The data are not publicly available.

**Acknowledgments:** We thank Tony Spiers and Select Harvests Ltd. for assistance and access to their orchards. We thank Joel Nichols, Nimanie Hapuarachchi and Anushika de Silva for laboratory assistance.

**Conflicts of Interest:** The authors declare no conflict of interest. The funder approved publication of the results but had no role in the design of the study; in the collection, analyses, or interpretation of data; in the writing of the manuscript.

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
