# Peer review of "Pollen Paternity Can Affect Kernel Size and Nutritional Composition of Self-Incompatible and New Self-Compatible Almond Cultivars"

_agronomy, doi:10.3390/agronomy11020326_

Round 1

Reviewer 1 Report

I think this submitted paper is well written. There is no problem, but the unit (m3; L104 and 202) need to correct.

Additional comments:

Most cultivars of almond are self-incompatible and require cross-pollination for fruiting. 

Xenia is a phenomenon in which pollinated pollen genes affect seed characteristics, which is a problem for crops that use seeds such as almond.

This paper investigates the characteristics of seeds produced by self- and cross- pollination in self-compatible cultivars of almond, which is highly original.

In addition, the results of investigating the ratio of seeds produced by self- and cross- pollination in almond produced by open-pollination by honeybees are very interesting.

Further investigations may be needed, but it may be possible to breed pollinizer that should be used for each cultivar by advancing this study.

 The unit of mass in Table 3 and 4 was not shown.    

Author Response

I think this submitted paper is well written. There is no problem, but the unit (m3; L104 and 202) need to correct.

Response: We corrected the unit and now use superscript at lines 103 and 204

‘The numbers of honey bees that contacted a flower within a 5-min period were count-ed in a 1 m3 quadrat on the illuminated side in both the morning (0800–1130 h) and the afternoon (1230–1600 h) of three days during peak flowering for seven cultivars.’

‘Honey bees were active in both orchards, with 4–11 bees observed in each 1 m3 quadrat in a 5-min window during peak flowering (Table 2).

The unit of mass in Table 3 and 4 was not shown.    

Response: We added the unit of mass for ‘in-shell mass’ and ‘kernel mass’ in Table 3 and 4.

Reviewer 2 Report

Dear authors, 
I found your work extremely interesting, not only from a scientific point of view, but also for the application. It is a very time-consuming and labour-intensive work. 

But the results are extremely interesting. 

I only ask you for a small revision: as also indicated by the journal guidelines , I ask you to remove from the keywords those already present in the title. 

In my opinion, if the manuscript is well done, it does not need much justification.
However, following the guidance of comments, I will give some details:  

What is the main question addressed by the research?

The main question addressed by research concerns the effects of the pollen paternity on quality traits of almonds. It is a very interesting topic for all nut crops.     

How original is the topic? What does it add to the subject area compared with other published material?

Even if the axenia is a well-known phenomenon, few papers have been published about this topics, since it requires very much work. Furthermore, the originality can be ascribed to the cultivars used. Until now, any published data reports information about axenia on these varieties    

Is the paper well written?As far as I know, the paper is well written. Anyway, I don't feel qualified because I'm not an english mother tongue.    

Is the text clear and easy to read?
The text is very clear and easy to read also for non scientific readers. 

    Are the conclusions consistent with the evidence and arguments presented? The conclusions are consistent and supported by strong data.  

Do they address the main question posed?
Yes, finally they address the main question posed. 

Author Response

COMMENTS FROM REVIEWER 2:

I only ask you for a small revision: as also indicated by the journal guidelines, I ask you to remove from the keywords those already present in the title. 

Response: We removed almond from the keywords and replaced self-compatible with self-fertile. The keywords listed are now:

‘Cross-pollination, fatty acid, kernel quality, nutrients, paternity, self-fertile’
